# Prospects of Continual Causality for Industrial Applications

**Daigo Fujiwara[1], Kazuki Koyama[1]\*, Keisuke Kiritoshi[1], Tomomi Okawachi[1],**
**Tomonori Izumitani[1], Shohei Shimizu[2]**

[1]NTT Communications Corporation, Japan
[2]Faculty of Data Science, Shiga University, Japan
{d.fujiwara, k.kiritoshi, t.okawachi, tomonori.izumitani}@ntt.com; shohei-shimizu@biwako.shiga-u.ac.jp

## Abstract

We have been investigating the causal analysis of industrial plant process data and its various applications, such as material quantity optimization utilizing intervention effects. However, process data often comes with various problems such as non-stationary characteristics including distribution shifts, which make such applications difficult. When combined with the idea of continual learning, causal models may be able to solve these problems. We present the potential and prospects for industrial applications of continual causality, showing previous work. We also briefly introduce our causal discovery method utilizing a continual framework.

## Our Position and Purpose

We have been researching business applications for industrial plants in which predictive models are created from data and the results are used for later actions to achieve specific objectives. Our findings have shown that the concepts of continual learning and causality are important for achieving these goals. Here, we present the challenges we have faced so far in terms of the combination of continual learning and causality (**continual causality**) and discuss potential solutions. We also briefly discuss a new causal discovery method to deal with non-stationarity and non-linearity by continual learning.

## Discussion

### Causality in Industrial Applications

In many industrial applications of AI, the purpose of a prediction is often to stabilize or maximize a specific variable using the predicted value, e.g., optimizing the output product for material input in an industrial plant. Since a simple prediction model may not accurately capture the data generation process, it can be difficult to estimate intervention effects, such as how much the production rate will increase when the material input is increased. Therefore, causal analysis is important for such applications. Causality is also useful from the viewpoint of interpretability because plants have a high risk of accidents and potential damage, which makes it important to understand the basis and reasons for various types of predictions.

While causality is useful, the complete picture of causal relationships is rarely available in plant process data. This is because plant processes often include feedback loops and material reuse, and there may be time-delayed effects among processes, resulting in causal relationships and directions that are often nontrivial. It is therefore vital to identify unknown causal relationships as well as to estimate intervention effects. However, it is difficult to conduct experiments such as Randomized Controlled Trials (RCTs) to identify causal relationships in non-operating conditions because of the risk of accidents, potential damage, and the various business factors in a given plant. This makes the framework of **causal discovery**, which identifies causal relationships and directions only from data, quite important.

## Potential of Continual Learning

To begin, in this paper, we broadly define Continual Learning as a framework in which new knowledge in machine learning should be learned in such a way that it reuses knowledge already learned and acquired, and can be learned continuously, hierarchically, and additionally. (Refer to this paper (Ring et al. 1994).) Note that our definition treats Continual Learning as something that is not necessarily strongly associated with reinforcement learning.

Modeling plant process data also runs into difficulties in terms of maintenance over time. For example, the average plant will exhibit many non-stationary characteristics, such as instability at start-up, distribution shifts due to changes in the quantity or type of products, trends related to equipment aging, and seasonality issues stemming from outdoor temperature changes (Kadlec, Gabrys, and Strandt 2009). By utilizing the concept of continual learning, the system can continuously train models and adapt to changes in system conditions. We have integrated this concept into a new method called JIT-LiNGAM (Fujiwara et al. submitted) (discussed later).

It may also be possible to reconsider the aforementioned tasks of stabilizing and maximizing plant process variables in the context of not only system control or causality but also reinforcement learning and continual learning (future work).

---

\*Current affiliation is School of Multidisciplinary Sciences, the Graduate University of Advanced Studies, SOKENDAI, Japan.

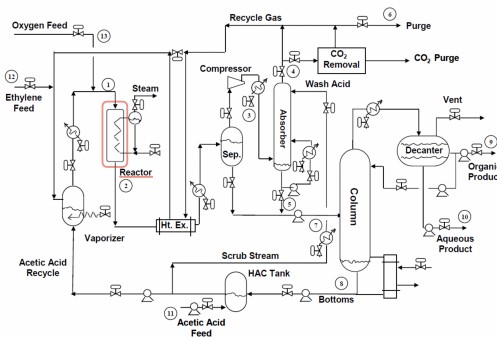

Figure 1: Flow of vinyl acetate production plant simulator (Luyben and Tyréus 1998).

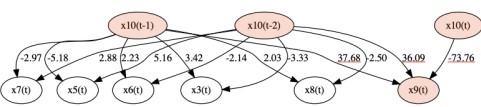

Figure 2: Results of applying VAR-LiNGAM to plant simulator data. Edges represent linear causality coefficients. Nodes x7–x10 denote process variables, e.g., x7(t-1) means the value of x7 one step before at time t.

## Past Efforts and Future Prospects

In this paper, we discuss the below challenges, describe the efforts of ourselves and other researchers to address them, and briefly mention future prospects.

### Causal Discovery

Causal discovery is a framework for identifying unknown causal relationships and directions only from data. As mentioned above, this framework is important because causal relationships are often unknown in plant process data. Discovered causal relationships are utilized for later intervention effect estimation and optimization, as well as for variable selection and model interpretation. LiNGAM (Shimizu et al. 2006) is a representative linear causal discovery method, and it has been extended with partial prior knowledge (Shimizu et al. 2011) and with latent variables (Hoyer et al. 2008). Several non-linear methods are also known (Peters et al. 2014; Zheng et al. 2020; Uemura et al. 2022).

We have applied these methods to actual plant process data but faced the common problem that the "true causal relationships" are unknown, which makes it difficult to evaluate the results. However, there is a possibility that the causal model can be continuously evaluated indirectly on the basis of the results of later interventional actions and then updated accordingly. This should be considered in future work related to continual causality.

### Time-series Extension

It is necessary to introduce time-series models to account for time-lagged variables in causal discovery, as this enables us to construct causal models without contradiction by expanding feedback loops along the time direction. Specific meth-

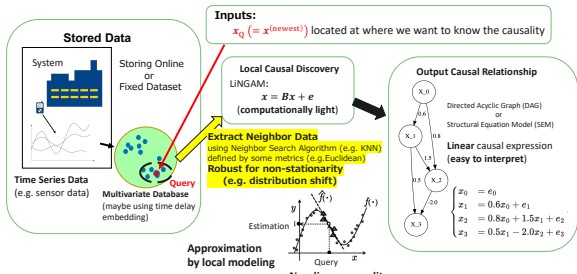

Figure 3: Overview of JIT-LiNGAM.

ods in this vein include VAR-LiNGAM (Hyvärinen et al. 2010). We have conducted numerical experiments in which VAR-LiNGAM is applied to the simulation data of vinyl acetate plants (Luyben and Tyréus 1998), with the results briefly presented in Figs. 1 and 2.

### Optimal Intervention

After constructing a complete causal model, the optimal amount of intervention to an operable variable can be calculated backward such that a certain variable has a specific value (Pearl, Glymour, and Jewell 2016). There are various extensions of this approach, including methods that utilize predictive models (Blöbaum and Shimizu 2017) or that estimate the optimal individual-level intervention (Kiritoshi et al. 2021).

### Continual Causal Discovery : JIT-LiNGAM

We proposed a causal discovery method for non-stationarity (e.g., distribution shift and non-linear causal relationships) called JIT-LiNGAM (Fujiwara et al. submitted), in which LiNGAM is combined with Just-In-Time Modeling (JIT) (Stenman, Gustafsson, and Ljung 1996; Bontempi, Birattari, and Bersini 1999). JIT is a method conventionally used for soft sensors (pseudo-sensors in plants for difficult-to-measure locations using regression models, etc.), where local linear models are trained continually by extracting neighboring samples of the current input sample from a database. On the basis of Taylor's theorem, non-linear phenomena in plants can be approximated by local linear models, and by utilizing neighboring samples for the modeling, we can follow continual changes in plants. The database can also be updated by adding samples online; however, due to limitations of memory and computational complexity, efficient use of data is vital. Future work should examine the optimal way of using data and consider the inclusion of other developments, such as the use of influence functions or of continual learning methods combined with reinforcement learning.

Extensions to time-delayed causality (as described above) and optimal intervention are also possible. In addition, since this approach enables us to capture snapshots of non-linear, non-stationary, and dynamically changing causal relationships, we may even be able to deal with cases where causal directions are being reversed. This is a potential solution to the plant feedback loop problem described above.

# Conclusion

We presented our positions in the causal analysis research area relevant to continual learning problems. We are currently working on each introduced theme independently, but in the future we will need to integrate them. In particular, we plan to extend JIT-LiNGAM in various ways. Continual causality is still a very much unexplored area, and extensive research will thus be conducted in the future.

# Appendix:JIT-LiNGAM Algorithm

We present the details of JIT-LiNGAM in Algorithm 1. Additional details are included in our submitted paper (Fujiwara et al. submitted).

---

**Algorithm 1: JIT Algorithm for Time-Series Causal Discovery (JIT-LiNGAM)**

---

**Inputs:**
    stored data $\mathcal{D} = \left\{ \boldsymbol{x}^{(t)} \mid t = 1, \ldots, T - 1 \right\}$,
    query point $\boldsymbol{x}_Q = \boldsymbol{x}^{(T)}$, distance function $d(\cdot, \cdot)$, number of neighbors $K$.

**Outputs:**
    weighted adjacency matrix $\boldsymbol{J}(\boldsymbol{x}^{(T)})$: representing the causality defined in the neighborhood for query point $\boldsymbol{x}^{(T)}$.

**Procedure 1**
Extract $K$-data of $\boldsymbol{x}^{(t)}$ from $\mathcal{D}$, on the basis of $d(\boldsymbol{x}^{(t)}, \boldsymbol{x}_Q)$, which is the distance from the query point $\boldsymbol{x}_Q$. (The details of how to extract $K$-data are described in the paper (Fujiwara et al. submitted)) The resulting $K$-data subset $\Omega(\boldsymbol{x}_Q; d, K)$ is:

$$\Omega(\boldsymbol{x}_Q; d, K) = \left\{ \boldsymbol{x}^{(\sigma(k))} \mid k = 1, \ldots, K \right\},$$

where $\sigma(k)$ is a function that returns the $k$-th nearest time index $t$ in $\Omega(\boldsymbol{x}_Q; d, K)$.

**Procedure 2**
Centralize $\Omega(\boldsymbol{x}_Q; d, K)$ and get $\tilde{\Omega}(\boldsymbol{x}_Q; d, K)$, where the mean is subtracted from each element of $\Omega(\boldsymbol{x}_Q; d, K)$ along each dimension of $\boldsymbol{x}$.

**Procedure 3**
Train LiNGAM using $\tilde{\Omega}(\boldsymbol{x}_Q; d, K)$ and get resulting weighted adjacency matrix $\boldsymbol{J}(\boldsymbol{x}^{(T)})$.

---

# Acknowledgments

I would like to thank Kazuki Koyama, Keisuke Kiritoshi, and Tomomi Okawachi for their contributions to experimental data acquisition and related research, and for helpful discussions. Special thanks also go to Tomonori Izumitani and Shohei Shimizu for their helpful guidance on the overall direction of the research and the writing of the paper in addition to the above-mentioned contributions. Finally, I would like to thank NTT Communications Corporation and its employees for the excellent research environment and support.

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
