# OpenReview forum: "Prospects of Continual Causality for Industrial Applications"
_AAAI.org/2023/Bridge/CCBridge — AAAI23 Bridge Continual Causality_

### Official Review · Reviewer_CgRw · 2022-12-01
**Interesting framework but difficult to evaluate due to brevity**

**Rating:** 4
**Confidence:** 4

**Review:**

- Applying the framework of continual causal learning to industry application sounds very interesting.

- However, I had trouble understanding what it is that the authors have done.

- I would have appreciated some more explanation of the figures in the paper.

- For work on extending causal models to include temporal information, you may find the following relevant Bramley et al. (2018).

References

Bramley, N. R., Gerstenberg, T., Mayrhofer, R., and Lagnado, D. A. (2018). Time in causal structure learning. Journal of Experimental Psychology: Learning, Memory, and Cognition, 44(12):1880--1910.

---

### Official Review · Reviewer_bS3e · 2022-12-02
**Causal model discovery in non-stationary Industrial settings**

**Rating:** 5
**Confidence:** 3

**Review:**

Summary: The authors propose to extend past work on causal model discovery in non-stationary settings involving industrial plant process monitoring.  The proposed work seems to be an incremental extension of past work where adaptations of LinGAM, a linear causal model discovery method, is incorporated along K-nn search.

Pros:  Time series analysis in industrial process monitoring settings is an application domain where continual causality will be of importance. The paper builds upon past work in the area.

Cons: The proposed work is incremental.

---

### Official Review · Reviewer_bMb2 · 2022-12-05
**Interesting ideas to be further explored**

**Rating:** 7
**Confidence:** 3

**Review:**

The paper discusses about causal analysis of industrial plants process data and its applications (e.g. material quantity optimization using intervention effects) in the context of non-stationary characteristics.

I found the extended abstract easy to read and clear, even though, at times, too informal. Potential and prospects for industrial applications of continual causality are interesting but should be better explored / motivated with a particular focus on the novel continual causal discovery method and how to put the two approaches together.

Overall, I found the paper quite aligned with the scope of the workshop and with some interesting suggestions for future research in the area.

---

### Official Review · Reviewer_eHMn · 2022-12-05
**The general position of why causal discovery for industrial applications of AI is discussed but the work does not outline why continual learning is necessary except for stating that there are distribution shifts and additionally fails to propose a synergy between continual learning and causal discovery, merely stating that continual learning is required without providing details or suggestions.**

**Rating:** 3
**Confidence:** 5

**Review:**

**Paper summary**: The position of the submitted work is that causality is essential to industrial applications of AI such as in industrial plant processing. Additionally, the industrial setting exposes the data and models to non-stationary characteristics such as distribution shifts which could be tackled with continual learning methods. The work recites past efforts on causal discovery such as LiNGAM (a linear causal discovery method) and VAR-LiNGAM (an extension of LiNGAM that introduces time-lagged variables for time series). To bridge the required continual aspect and the causal discovery, the work proposes JIT-LiNGAM which combines LiNGAM with JIT (just-in-time modeling).

While the need for causal models and the suggestion of using LiNGAM for causal discovery in industrial applications is outlined adequately, the continual learning aspect seems to be mostly missing. The work describes the necessity for continual methods due to distribution shifts in various processes but fails to specify how to link the two fields. It is unclear, what the time-series extension to LiNGAM (VAR-LiNGAM), the paragraph on optimal intervention, and the proposed JIT-LiNGAM for dynamical systems has to do with continual learning. What is the continual learning aspect of the proposed method? The work only mentions that this should be combined with continual learning (e.g. in the &ldquo;Potential of Continual Learning&rdquo; paragraph &ldquo;*By using the concept of continual learning, the system can be expected to continuously train models and adapt to changes in system conditions*&rdquo; and at the end of the JIT-LiNGAM paragraph: &ldquo;*or utilizing continual learning methods and reinforcement learning, may be considered.*&rdquo;). The description of JIT-LiNGAM mentions, that &ldquo;*local linear models are trained continually*&rdquo; but omits what this even means. Unfortunately, the work wastes precious space by including figures 1 and 2 which show a production flow of vinyl acetate and the results of applying VAR-LiNGAM to plant simulator data. The relevance of the figures to the proposed method is not discussed in the text.

---

### Decision · Program_Chairs · 2022-12-05

**Decision:**

Accept

**Comment:**

Accept - Poster

This paper posits that causality is essential to applications of industrial plant processes. While the contributions may be limited, the application is interesting and well aligned with the bridge program. Therefore, we accept this paper but strongly encourage the authors to read the reviewer comments and feedback and address them using the extra page allowed in the camera-ready version of the paper. In particular, we would like the authors to clarify 1) their scientific contributions and 2) the link between causality and continual learning.